# Assessing the Sensitivity of Multi-Distance Hyperspectral NIRS to Changes in the Oxidation State of Cytochrome C Oxidase in the Brain

**DOI:** 10.3390/metabo12090817

**Published:** 2022-08-31

**Authors:** Marianne Suwalski, Leena N. Shoemaker, J. Kevin Shoemaker, Mamadou Diop, John M. Murkin, Jason Chui, Keith St. Lawrence, Daniel Milej

**Affiliations:** 1Department of Medical Biophysics, Western University, 1151 Richmond St, London, ON N6A 3K7, Canada; 2Imaging Division, Lawson Health Research Institute, Imaging Program, 268 Grosvenor St, London, ON N6A 4V2, Canada; 3Department of Kinesiology, Western University, 1151 Richmond St, London, ON N6A 3K7, Canada; 4Department of Anesthesiology and Perioperative Medicine, London Health Science Centre, 339 Windermere Rd, London, ON N6A 5A5, Canada

**Keywords:** cytochrome c oxidase, hyperspectral NIRS, carotid compression, tissue oxygen saturation, diffuse correlation spectroscopy, blood flow index, hypercapnia

## Abstract

Near-infrared spectroscopy (NIRS) measurements of tissue oxygen saturation (StO_2_) are frequently used during vascular and cardiac surgeries as a non-invasive means of assessing brain health; however, signal contamination from extracerebral tissues remains a concern. As an alternative, hyperspectral (hs)NIRS can be used to measure changes in the oxidation state of cytochrome c oxidase (ΔoxCCO), which provides greater sensitivity to the brain given its higher mitochondrial concentration versus the scalp. The purpose of this study was to evaluate the depth sensitivity of the oxCCO signal to changes occurring in the brain and extracerebral tissue components. The oxCCO assessment was conducted using multi-distance hsNIRS (source-detector separations = 1 and 3 cm), and metabolic changes were compared to changes in StO_2_. Ten participants were monitored using an in-house system combining hsNIRS and diffuse correlation spectroscopy (DCS). Data were acquired during carotid compression (CC) to reduce blood flow and hypercapnia to increase flow. Reducing blood flow by CC resulted in a significant decrease in oxCCO measured at *r*_SD_ = 3 cm but not at 1 cm. In contrast, significant changes in StO_2_ were found at both distances. Hypercapnia caused significant increases in StO_2_ and oxCCO at *r*_SD_ = 3 cm, but not at 1 cm. Extracerebral contamination resulted in elevated StO_2_ but not oxCCO after hypercapnia, which was significantly reduced by applying regression analysis. This study demonstrated that oxCCO was less sensitive to extracerebral signals than StO_2_.

## 1. Introduction

Various procedures employed during cardiac and vascular surgery, such as cardiopulmonary bypass or arterial clamping, place the patient at risk of brain injury, with an incidence of postoperative stroke between 0.8 and 5.2% [1,2,3,4] and cognitive decline between 14.1 and 50% [5,6]. In an effort to reduce the risk of neurological complications, brain monitoring has become an essential component of intraoperative management. Several techniques have been evaluated, including electroencephalography (EEG) [7], somatosensory evoked potential (SEP) [8], transcranial Doppler (TCD) [9], and cerebral tissue oxygen saturation (StO_2_) by near-infrared spectroscopy (NIRS) [10]. A disadvantage of EEG and SEP is that the signals only indirectly reflect cerebral blood flow (CBF) [11]. While a decrease in amplitude in EEG or SEP can indicate reduced CBF, not all EEG and SEP changes are associated with ischemic injury, and stroke can occur even in the absence of changes [12]. Unnecessary shunt placement during carotid endarterectomies is associated with TCD monitoring since only changes in the mean blood velocity in the major conduit arteries are measured, which may not reflect changes in the microvasculature blood flow [13].

The clinical applications of commercially available NIRS systems continue to grow, given their ability to monitor StO_2_ non-invasively by detecting changes in oxy- (HbO_2_) and deoxyhemoglobin (Hb) concentrations. However, a well-known challenge with monitoring StO_2_ is substantial signal contamination from extracerebral tissues [14,15]. Different algorithms have been developed to improve brain sensitivity, but assessing cerebral StO_2_ accurately remains challenging [13,16,17]. Furthermore, StO_2_ is not a direct marker of CBF or cerebral oxygen demands.

In contrast to StO_2_, cytochrome c oxidase (CCO) is a proton pump that plays a vital role in producing energy, in the form of ATP through oxygen metabolism [18,19,20]. Since CCO accounts for 95% of the total uptake of O_2_, as long as there are no changes in the supply of electrons from substrates (i.e., NADH) or in the concentration of CCO, and no terminal inhibitors (i.e., NO), changes in oxygen availability will result in changes in the oxidation state of CCO (ΔoxCCO) [21].Changes in the oxidation state of one of CCO’s centers, Copper A, are reflected in absorption changes in the NIR range [22]. Therefore, measuring ΔoxCCO has the potential to be used as a marker of brain health, especially in clinical scenarios. A further advantage of monitoring oxCCO [23,24], rather than StO_2_, is its greater brain sensitivity due to the higher mitochondrial concentration in the metabolically active brain compared to the scalp [25,26].

Despite these advantages, measuring ΔoxCCO is challenging since the absorption features of oxCCO in the NIR spectrum are broad, and its concentration is less than 10% of hemoglobin [19]. It has been shown that assessing ΔoxCCO reliably requires measuring absorption changes across many wavelengths, which is typically performed using broadband or hyperspectral (hs) NIRS. We previously demonstrated that hsNIRS could detect changes in ΔoxCCO during cardiac surgery with cardiopulmonary bypass and how ΔoxCCO reacted differently from CBF and StO_2_, indicative of its intrinsic sensitivity to metabolism [27]. However, these previous studies were conducted using a hsNIRS system with a single source-detector separation, and therefore, despite the greater brain sensitivity of oxCCO, possible signal contributions from the scalp were unknown [28].

The objective of the current study was to assess signal contributions from the scalp and brain by acquiring hsNIRS data at two source-detector distances (i.e., *r*_SD_ = 1 and 3 cm). Experiments were conducted on healthy adult subjects and involved two stimuli chosen to reflect both reduced and excessive blood flow that can occur during cardiac and vascular surgery: carotid compressions (CC) [29,30,31] and hypercapnia [28,32,33]. In addition, the signal measured at *r*_SD_ = 1 cm was used as a regressor to reduce extracerebral contributions from the data acquired at *r*_SD_ = 3 cm [34,35]. All experiments were conducted using an in-house built hybrid hsNIRS/diffuse correlation spectroscopy (DCS) neuromonitoring system modified to collect broadband NIR spectra from two distances. This hybrid system also enabled concurrent monitoring of a blood flow index (BFi) from DCS [27,36].

## 2. Methods

### 2.1. Instrumentation

The hsNIRS light source was a 20-W halogen bulb (Hl-2000-HP, Ocean Optics, Largo, FL, USA) that was filtered from 600 to 1000 nm and coupled into a custom optical fiber bundle (Ø = 2.4 mm, Φ *=* 30 μM, NA = 0.55; Loptek, Germany) that directed the light to the subject’s head. The system incorporates two spectrometers to acquire absorption spectra at *r*_SD_ = 1 and 3 cm. At *r*_SD_ = 1 cm, reflected light was collected by a multimode optical fiber (Φ *=* 600 μm, NA = 0.39; Thorlabs, Newton, NJ, USA) and transmitted through a shutter to a spectrometer (AvaSpec-ULS2048XL, λ_Bandwith_ = 666–1025 nm, λ_Resolution_ = 0.18 nm; Avantes, The Netherlands). At *r*_SD_ = 3 cm, reflected light was collected by three fiber bundles (Ø = 2 mm, Φ *=* 30 μm, NA = 0.55; Loptek, Germany) that were linearly aligned at the entrance of a second spectrometer (iDus 420, λ_Bandwith_ = 548–1081 nm, λ_Resolution_ = 0.52 nm; Andor, Oxford Instruments, Toronto, ON, Canada). Reflectance spectra were acquired at both distances simultaneously.

For the DCS module, light from a long coherence laser (DL785-100s, CrystaLaser, Reno, NV, USA) was coupled into a fiber bundle (Φ *=* 4 × 200 μm, NA = 0.22; Loptek, Berlin, Germany). The reflected light was collected by four single-mode fibers (Φ = 8 μm, NA = 0.12; Loptek, Berlin, Germany) and coupled to a four-channel single photon counting module (SPCM-AQR-15-FC, Excelitas Technologies, Toronto, ON, Canada). Each counting module generated TTL pulses that were sent to an edge-detecting photon counter on a PCIe6612 counter/timer data acquisition board (National Instrument, Austin, TX, United States) [37,38]. Photon counts were recorded (LabVIEW 2017 SP1, National Instruments, USA) and processed using in-house developed software (MATLAB 2016b, MathWorks, Natick, MA, USA). For each detector, the software generated intensity autocorrelation curves at 50 delay times (*τ*) ranging from 1 μs to 1 ms [14,37].

### 2.2. Experimental Protocol

All experiments were approved by the Western University Health Sciences Research Ethics Board, which adheres to the guidelines of the Tri-Council Policy Statement for research involving humans. Written informed consent was obtained from each participant before the experiment. Volunteers were excluded based on a neurological or psychiatric disorder diagnosis or a history of vascular disease. All participants completed both the carotid compression and hypercapnia experiments.

Participants were studied in the supine position. Optical probes were attached to the right side of their forehead via a custom-designed probe holder secured by a Velcro headband. One detection fiber bundle was placed at *r*_SD_ = 1 cm, and three detection fiber bundles, which collected both NIRS and DCS signals, were placed at *r*_SD_ = 3 cm from the NIRS source and 2 cm from the DCS source (Figure 1), respectively.

Continuous arterial blood pressure was measured by finger photoplethysmography (Finometer, Finapres Medical Systems, Enschede, Netherlands), which was calibrated against three manual measurements for the sphygmomanometric brachial artery. Arterial blood pressure was used to calculate mean arterial pressure (MAP).

#### 2.2.1. Carotid Compressions (CC)

The experimental protocol for CC consisted of a 1-min baseline period followed by three digital compressions of the right (i.e., ipsilateral to the position of the probes) common carotid artery at the level of 1 cm superior to the clavicle (Figure 2) [39]. Each compression lasted for 15 s, followed by a 30-s recovery period. The procedure was then performed on the left common carotid artery (i.e., contralateral to the position of the probes). Finally, compression was repeated on the right common carotid artery for a single 30-s period, followed by 1.5 min of recovery.

The hsNIRS/DCS system enables hsNIRS and DCS data to be collected sequentially using a multiplexing shutter system; however, due to the rapid response to CC, hsNIRS and DCS data were collected separately during these experiments.

#### 2.2.2. Hypercapnia

Subjects were required to wear a facemask connected to a computer-controlled gas delivery circuit (RespirAct, Thornhill Research Inc, Toronto, ON, Canada). The experimental protocol consisted of one 4-min period of hypercapnia in which end-tidal carbon dioxide pressure (P_ET_CO_2_) was increased by 10 mmHg above each subject’s normocapnic P_ET_CO_2_ value, as determined by the gas delivery circuit. The hypercapnic period started two minutes after baseline recordings and was followed by three minutes of normocapnia. Hyperspectral NIRS and DCS data were recorded sequentially during the experiment.

### 2.3. Data Analysis

#### 2.3.1. Hyperspectral NIRS

At the beginning of each experiment [40], a reference spectrum (*reference_λ_*) and a dark count spectrum (*dark_λ_*) were acquired for each spectrometer (i.e., one at *r*_SD_ = 1 cm and the other at *r*_SD_ = 3 cm). Spectra (*data_λ_*) collected during the baseline period before either CC or hypercapnia were converted into baseline reflectance spectra using the following:(1)$$R\left(\lambda \right)=log_{10}\left(\frac{data_{\lambda }-dark_{\lambda }}{reference_{\lambda }-dark_{\lambda }}\right)$$

The first and second derivatives of *R*(*λ*) were fitted with the solution to the diffusion approximation for a semi-infinite homogeneous medium [41] to generate estimates of the tissue water fraction, HbO_2_ and Hb concentrations, and two scattering parameters (wavelength-dependent power and the reduced scattering coefficient (*µ*_s_′) at 800 nm) [36]. Fitting was performed using a constrained minimization algorithm based on the MATLAB function fminsearchbnd (2016b, MathWorks, USA). The HbO_2_ and Hb concentrations estimates were used to calculate baseline tissue oxygen saturation at *r*_SD_ = 1 and 3 cm.

Changes in Hb, HbO_2_, and oxCCO concentrations relative to their baseline values were estimated using the modified Beer–Lambert law based on the UCLn algorithm [18]. The analysis was performed separately for spectra acquired at *r*_SD_ = 1 and 3 cm. Changes in Hb and HbO_2_ concentrations were determined from attenuation changes measured between *λ* = 680 and 850 nm [42]. Likewise, changes in oxCCO concentration were determined from attenuation changes between *λ* = 770 and 906 nm. For this analysis, the differential pathlength for each subject was calculated by fitting the second derivative of average baseline *R*(*λ*) to the second derivative of the water absorption spectrum [43] and correcting for the wavelength dependence of the pathlength [44]. StO_2_ at each time point was determined by combining the relative changes derived from the UCLn algorithm with the absolute baseline value obtained by derivative spectroscopy. The StO_2_ time courses were smoothed with a zero-phase digital filter (filtfilt, MATLAB, 2016b, MathWorks, Natick, MA, USA).

#### 2.3.2. DCS

Using the Siegert relation, normalized intensity autocorrelations functions were converted to electric field autocorrelation data [45]. Each autocorrelation function was subsequently fit with the diffusion approximation solution for a semi-infinite homogenous medium. The fitting incorporated assumed values of the optical coefficients *µ*_a_ = 0.1 cm^−1^ and *µ*_s_′ = 10 cm^−1^ [46] and the coherence factor (*β*) determined from the average initial value of the baseline g_2_ curves. The fitting procedure was performed across all correlation times from 1 μs to 1 ms and yielded a best-fit estimate of the blood flow index (BFi) based on modelling tissue perfusion as pseudo-Brownian motion [47]. The resulting BFi time courses were smoothed with the same filter applied to the hsNIRS data (i.e., zero-phase digital filtering; filtfilt, MATLAB, 2016b, MathWorks, USA).

#### 2.3.3. Regression Analysis

Regression analysis was described in detail previously [35]. It is based on the method proposed by Saager et al. [34] developed to isolate absorption trends in the lower layer of a two-layer turbid medium. The oxCCO and StO_2_ signal changes in the brain layer (i.e., ΔoxCCO_Reg_ and ΔStO_2,Reg_, respectively) were calculated according to ΔoxCCO_Reg_ = ΔoxCCO_3cm_ − α·ΔoxCCO_1cm_ and ΔStO_2,Reg_ = ΔStO_2,3cm_ − α·ΔStO_2,1cm_, where α is the scaling parameter obtained by using a least-squares criterion to fit the time series recorded at *r*_SD_ = 1 cm to the corresponding time series recorded at *r*_SD_ = 3 cm, (i.e., ΔoxCCO_1cm_ − ΔoxCCO_3cm_ and ΔStO_2,1cm_ − ΔStO_2,3cm_, where ΔoxCCO_1cm_ is the ΔoxCCO at *r*_SD_ = 1 cm, ΔoxCCO_3cm_ is ΔoxCCO at *r*_SD_ = 3 cm, etc.).

#### 2.3.4. Cerebrovascular Reactivity

To determine the response time of ΔBFi, ΔStO_2_, and ΔoxCCO to 30-s CC, the time courses of ΔBFi, ΔStO_2_, and ΔoxCCO were modelled as the convolution of a step function representing carotid compression (denoted *CC*(*t*) and scaled to a maximum value of one) and a hemodynamic response function (*HRF*) [33,48]:(2)ΔS(t)=ssCVR [CC(t)∗HRF(t)] 
where ∆*S*(*t*) is the signal change, *ssCVR* is the steady-state value of cerebrovascular reactivity (CVR) and * denotes the convolution operator. The *HRF* is given by:(3)$$HRF\left(t\right)=\left(\frac{1}{N}\;\right)e^{\frac{-\left(t-t_{0}\right)}{\tau }}\;$$
where *τ* is the time constant defining the dynamic component of CVR, $t_{0}$ is the time delay between the start of CC(t) and the initial decline of ∆*S*(*t*), and *N* is the area under $\int_{0}^{\infty }e^{-t/\tau }dt$. Best-fit estimates of *τ*, $t_{0}$, and *ssCVR* were obtained by numerical optimization (fminsearchbnd, MATLAB, Mathworks Inc., USA). The fitting was performed over a time window spanning the beginning of CC to the nadir of ∆*S*(*t*). For ΔStO_2_ and ΔoxCCO, the analysis was performed for time courses recorded at *r*_SD_ = 3 cm.

#### 2.3.5. Statistical Analysis

All data are presented as mean ± standard deviation unless otherwise noted. Statistical analyses were conducted in IBM SPSS. Statistical significance was defined as *p* < 0.05. Multivariate analyses of variance (ANOVA) were used to compare ΔStO_2_ and ΔoxCCO at the two *r*_SD_ (1 and 3 cm) during the two compression durations (15 and 30 s). Independent-samples *t*-tests were used to evaluate ΔStO_2_ and ΔoxCCO at the two *r*_SD_ (1 and 3 cm) and ΔStO_2.3cm_ and ΔoxCCO_3cm_ versus ΔStO_2_,_Reg_ and ΔoxCCO_Reg_. Paired-samples *t*-tests were used to evaluate ΔStO_2,1cm_, ΔoxCCO_1cm_, ΔBFi, and change in MAP versus the baseline. A repeated measures ANOVA was conducted on the 15-s ipsilateral CC data to determine the precision of ∆oxCCO and ∆StO_2_. Precision was defined by the coefficient of variation (CoV) for the within-subject variance.

## 3. Results

Data were acquired from 10 participants (4 females, 6 males, 24–34 years, mean = 29 ± 5 years). A total of 67 digital common carotid artery compressions were performed (30 15-s right CCs, 28 15-s left CCs, and 10 30-s right CCs). Data from one participant were excluded from the contralateral 15-s CC analysis as the participant experienced mild syncopal symptoms during the contralateral 15-s CC. The same 10 participants also underwent the hypercapnia protocol.

### 3.1. Carotid Compressions (CC)

Figure 3 presents time courses of average changes in BFi, StO_2_, and oxCCO in response to ipsilateral CC across subjects during the two compression durations. Data for ΔStO_2_ and ΔoxCCO are presented for both source-detector separations (*r*_SD_ = 1 and 3 cm). Decreases in ΔBFi, ΔStO_2_, and ΔoxCCO were observed at both source-detector distances during ipsilateral 15 s and 30 s CC. Change in each parameter in response to CC was characterized by taking the average of 5 s around the maximum reduction (Table 1).

Thirty-second CC resulted in a significant decrease in BFi (−57 ± 14%) and an increase in MAP (4 ± 1 mmHg). Decreases in oxCCO recorded at *r*_SD_ = 3 cm (Table 1) were significantly larger than the reductions measured at 1 cm (−0.06 ± 0.1 μM). The corresponding decrease in StO_2_ at the longer *r*_SD_ (−4 ± 2.2%) was also significantly larger than the decrease recorded at *r*_SD_ = 1 cm. The reduction in StO_2_ recorded at 1 cm was significantly less than baseline, whereas the reduction in oxCCO at 1 cm did not reach significance.

The 15-s CC response was averaged over the three trials for every subject. The average significant decrease in BFi was 55 ± 8%, and an increase in MAP (4 ± 3 mmHg). Similar to 30-s CC, reductions in ΔoxCCO and ΔStO_2_ recorded at *r*_SD_ = 3 cm were significantly greater than the corresponding reductions measured at 1 cm. StO_2_ and oxCCO changes measured for 15-s CC were not statistically different from those obtained for the 30-s CC. From the three 15-s CC trials, the estimated CoV for within-subject variability was 6% and 1% for ΔoxCCO at *r*_SD_ = 1 and 3 cm, respectively. Similar values were found for the corresponding ΔStO_2_ measurements: CoV = 9% and 8% at *r*_SD_ = 1 and 3 cm, respectively.

Fitting the cerebrovascular reactivity model to the time courses for 30-s CC demonstrated that ΔBFi exhibited the fastest response and ΔStO_2_ the slowest, as indicated by the time constant defining HRF(t); i.e., *τ* = 1.8 ± 1.4 s for ΔBFi, 4.8 ± 3.5 s for ΔoxCCO, and 14.8 ± 8.4 s for ΔStO_2_. The average *τ* value for ΔStO_2_ was significantly different from the corresponding values for ΔBFi and ΔoxCCO, while the values for ΔoxCCO and ΔBFi were not significantly different from each other. Despite the lack of significant difference between the response time between ΔBFi and ΔoxCCO, an average temporal delay of 4.7 ± 7.3 s was found between the nadirs.

Following completion of carotid compression, a brief 5-s hyperemic response was observed, which was characterized by a BFi increase of 32 ± 20% after 30-s CC and 28 ± 26% after 15-s CC; however, there was no significant change in ΔStO_2_ and ΔoxCCO.

Figure 4 presents average time courses of ΔStO_2_ and ΔoxCCO measured at *r*_SD_ = 3 cm in response to 30-s CC after applying regression analysis. In both cases, regression reduced the magnitude of the response to CC. Both reductions remained significantly different from baseline after regression. The maximum decrease was 2.4 ± 1.9% for ΔStO_2_ (Figure 4a) and 0.21 ± 0.24 µM for ΔoxCCO (Figure 4b). Regression also significantly reduced the time constant for regressed ΔStO_2_ (*τ* = 6.6 ± 5.6 s), and the average *τ* value for regressed ΔStO_2_ was not significantly different from the *τ* values for ΔBFi and ΔoxCCO.

Figure 5 presents the correlation of ΔoxCCO to ∆BFi during CC. A strong non-linear relationship can be observed for ΔoxCCO recorded at *r*_SD_ = 3 cm, with all ΔoxCCO values for BFi ≥ 21% significantly different from zero. In contrast, ΔoxCCO recorded at *r*_SD_ at 1 cm was relatively unresponsive to ∆BFi, with no values significantly different from zero.

Small decreases in ΔStO_2_ and ΔoxCCO were observed on the contralateral hemisphere in response to CC of 15 s (Figure 6); however, these changes did not reach significance. In contrast, a significant increase in BFi (14 ± 14%) was found.

### 3.2. Hypercapnia

Figure 7 presents average time courses of changes in BFi, StO_2_, and oxCCO in response to 4 min of hypercapnia (P_ET_CO_2_ increase = 10 ± 2 mmHg). Hypercapnic responses were calculated as the relative difference between the average signal from 4 to 6 min (i.e., 2nd half of the hypercapnic period) and the average of the first minute of baseline. Average ΔoxCCO and ΔStO_2_ recorded at both source-detectors distances are provided in Table 2. ΔoxCCO and ΔStO_2_ measured at *r*_SD_ = 3 cm were significantly larger than the responses measured at *r*_SD_ = 1 cm. The ΔoxCCO and ΔStO_2_ responses at 1 cm were significantly delayed (33 ± 24 s and 10 ± 6 s, respectively) compared to the corresponding responses at 3 cm. Hypercapnia resulted in a significant increase in BFi (31 ± 48%) and MAP (4 ± 1 mmHg). Persistent signal changes were observed after hypercapnia. These were compared to both baseline and hypercapnia by taking the average of the signal from 7 to 9 min. Only the post-hypercapnia ΔoxCCO measured at *r*_SD_ = 3 cm was significantly lower than its corresponding hypercapnic value (Table 2).

Figure 8 displays time-varying changes in oxCCO and StO_2_ averaged across subjects in response to hypercapnia after regression analysis. Regression reduced the magnitude of the hypercapnic increases for both oxCCO and StO_2_ (Table 2); however, their responses were not significantly different from the original responses measured at *r*_SD_ = 3 cm. Post-hypercapnia, ΔStO_2,Reg_ returned to baseline and was significantly smaller than the corresponding post-hypercapnia ΔStO_2,3cm_.

## 4. Discussion

This study focused on evaluating contributions from the scalp and brain on metabolic and hemodynamic markers measured with hsNIRS. The primary motivation was to improve the confidence in non-invasive ΔoxCCO monitoring for cardiac and vascular surgery applications. In this study, the impact of the scalp was assessed by comparing signals measured at *r*_SD_ = 1 cm, which predominately represents changes in the extracerebral layer, and *r*_SD_ = 3 cm, which contains a greater brain contribution. The study involved two paradigms: unilateral CC and hypercapnia. The motivation for using CC was that it is a safe method of causing rapid and large decreases in cerebral blood flow that mimics arterial occlusion performed during surgery. Hypercapnia was included given its well-known vasodilatory effects in the brain.

The average reductions in BFi for 15 and 30-s periods of CC were 55 ± 8% and 57 ± 14%, respectively, which are consistent with a 60% decrease in mean blood flow velocity measured in the middle cerebral artery by transcranial Doppler [30]. Repeat 15-s CC trials demonstrated that ΔoxCCO measured at both source-detector distances was highly reproducible with a CoV of 6% at *r*_SD_ = 1 cm and 1% at 3 cm. Thirty seconds of CC decreased oxCCO by 0.4 ± 0.3 µM at *r*_SD_ = 3 cm (Figure 3a). The magnitude of this decrease is greater than reported for other experimental paradigms, including mild hypoxia, hypocapnia [28], and breath holding [49]. More importantly, the average oxCCO reduction at *r*_SD_ = 3 cm was almost seven times greater than the corresponding oxCCO reduction measured at *r*_SD_ = 1 cm (0.06 ± 0.1 μM). The latter was not significantly different from the baseline. The significant difference in the oxCCO responses at the two distances (*p* = 0.012, ΔoxCCO at *r*_SD_ = 3 vs. 1 cm) reflects the greater brain contribution to the signal measured at *r*_SD_ = 3 cm. Considering the higher metabolic rate of the brain compared to scalp and the higher cerebral oxCCO concentration, a sudden and sizable decrease in oxygen delivery would likely have a greater effect on the brain. The greater sensitivity of the oxCCO signal to the brain is exemplified in Figure 5, which shows that changes in oxCCO measured at *r*_SD_ = 1 cm never reached significance across all BFi decreases; whereas, changes in oxCCO at *r*_SD_ = 3 cm were significant for all decreases in BFi greater than 20%.

The hypercapnia results also demonstrated the sensitivity of the oxCCO signal to the brain. The average increase in P_ET_CO_2_ was 10 ± 2 mmHg, which caused a 31 ± 48% increase in BFi and a significant increase in oxCCO of 0.22 ± 0.19 μM measured at *r*_SD_ = 3 cm (Table 2). Similar to the CC results, the oxCCO change measured at *r*_SD_ = 1 cm (0.1 ± 0.1 μM) did not reach significance. This finding is in agreement with Kolyva et al., who reported that the magnitude of the oxCCO response to hypercapnia increased with source-detector separation (2 ≤ *r*_SD_ ≤ 3.5 cm) [32]. Note, there is some debate as to whether oxCCO should increase during hypercapnia if it is close to fully oxidized at normoxia [50,51]. The consistent increase in oxCCO observed in human participants indicates that this is likely not the case [32].

This study also demonstrated that StO_2_ was more sensitive to extracerebral tissue than oxCCO. Similar to oxCCO, greater changes in StO_2_ were measured at *r*_SD_ = 3 cm compared to 1 cm for CC; however, the ratio of ΔStO_2_ measured at the two distances was around three, in contrast to a ratio closer to seven for ΔoxCCO. Moreover, unlike oxCCO, there was a significant decrease in StO_2_ measured at *r*_SD_ = 1 cm (*p* = 0.001 vs. baseline) (Table 1). The hemodynamic response of ΔStO_2_ to CC was also significantly slower, as characterized by the time constant *τ*, which was larger for ΔStO_2_ compared to the corresponding values for ∆oxCCO and ∆BFi. The average time courses for the three parameters (Figure 3) demonstrated that ΔoxCCO followed ∆BFi more closely than ΔStO_2_. The StO_2_ response likely reflects a slower response to CC in the metabolically inactive scalp tissue. In newborn piglets, which have thin skulls and negligible scalp muscle, Rajaram et al. observed that CBF and StO_2_ both decreased rapidly in response to hypoxia-ischemia while oxCCO displayed a delayed response [40]. The use of hypoxia in the piglet study may also have contributed to the difference between these two studies since SaO_2_ was not altered in the CC experiments.

A further illustration of the sensitivity of StO_2_ to the extracerebral tissue was the persistent elevation observed after hypercapnia (Figure 7). In healthy participants, cerebrovascular reactivity will be reflected by rapid changes in StO_2_ at the onset and end of hypercapnia, as demonstrated in functional magnetic resonance imaging studies [48]. The influence of the scalp, which has considerably more sluggish vascular reactivity, was previously demonstrated using time-resolved NIRS. Only hemoglobin signals with enhanced depth sensitivity exhibited a rapid return to baseline when P_ET_CO_2_ returned to normocapnia [33,35]. In the current study, ΔStO_2_ at *r*_SD_ = 3 cm remained significantly greater than baseline (*p* = 0.003) one to three minutes after hypercapnia. In contrast, ΔoxCCO at *r*_SD_ = 3 cm in the same period was not significantly different from baseline. However, some evidence of scalp contamination in oxCCO measurements was also observed. Post hypercapnia, ΔoxCCO at *r*_SD_ = 1 cm was significantly greater than at baseline, reflecting some sensitivity to the scalp (Table 2).

Regression analysis was explored as a means of reducing scalp contamination in the hsNIRS data. For both ΔoxCCO and ΔStO_2_ during CC, regression reduced the magnitude and inter-subject variability (Figure 4); however, these changes were not significant. More apparent effects can be observed in the regression results obtained for hypercapnia (Figure 8). For the hypercapnia data, regression did significantly reduce post hypercapnia ΔStO_2_ (*p* = 0.02 vs. ΔStO_2_ at 3 cm) (Table 2). A similar effect can be observed for ΔoxCCO; however, the signal change did not reach significance. This effect suggests that scalp contamination also affected the ΔoxCCO signal at *r*_SD_ = 3 cm but to a lesser extent than ΔStO_2_.

This study presented a few limitations. The ratio of the CC responses at the two source-detector separations (Table 1) was larger for ΔoxCCO than StO_2_, but this difference was not significant. Power analysis indicated that 42 participants would have been required to show significance. Next, only a single source-detector separation was used for DCS acquisition. While the DCS will include contributions from scalp blood flow, the magnitude of this contamination will be less compared to NIRS due to the higher blood flow in the brain. Since this study was primarily focused on hsNIRS measurements of oxCCO, single-distance DCS measurements were deemed reasonable. Finally, regression analysis is sensitive to signal noise and cannot be performed in real-time in clinical settings. Future work will focus on assessing real-time methods for reducing scalp contributions from multi-distance hsNIRS data.

## 5. Conclusions

In summary, the study measured oxCCO and StO_2_ changes with hsNIRS at multiple source-detector distances during two paradigms. The first paradigm, CC, caused substantial reductions in blood flow, analogous to hemodynamic events that can occur during cardiac and vascular surgeries. The novelty of this study was demonstrating a significant decrease in oxCCO at *r*_SD_ = 3 cm during CC but not at *r*_SD_ = 1 cm. In contrast, significant decreases in StO_2_ were observed at both distances. These results indicate that oxCCO had less scalp contamination than concurrent StO_2_ measurements. These results highlight the potential of using oxCCO to monitor brain health during surgery. However, increases in oxCCO were observed in the post hypercapnia data acquired at *r*_SD_ = 1 cm, indicating some contamination from the scalp. Therefore, acquiring multi-distance hsNIRS data and applying methods to separate scalp and brain contributions, such as regression analysis, is likely prudent for interpreting changes in oxCCO in clinical settings.

## Figures and Tables

**Figure 1 metabolites-12-00817-f001:**
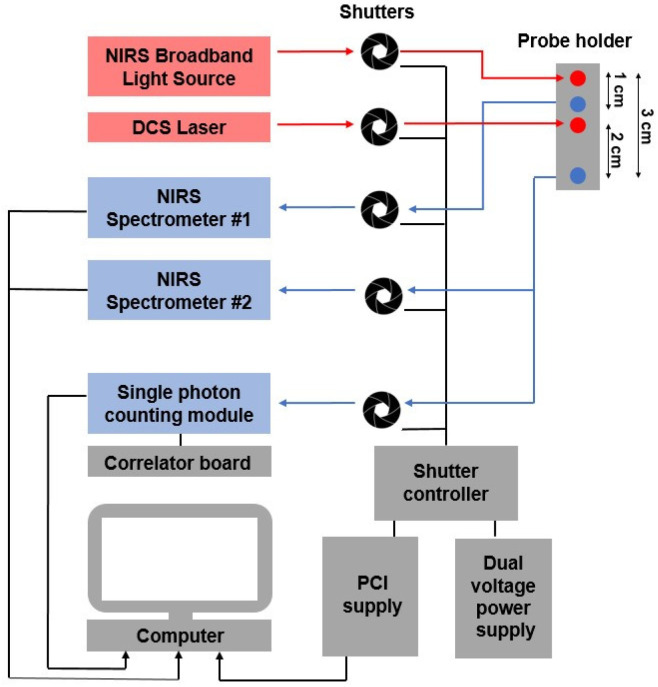
Schematic of the hsNIRS/DCS system and optical probe holder.

**Figure 2 metabolites-12-00817-f002:**
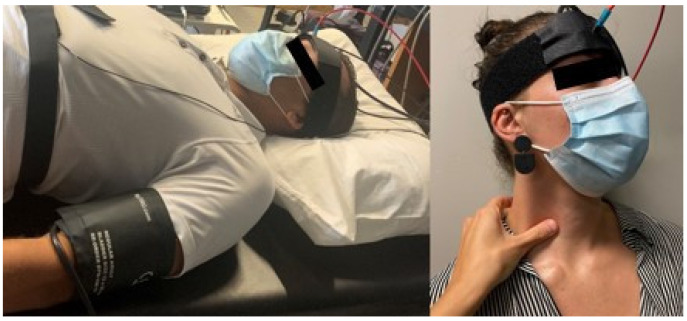
Experimental participant set-up and carotid compression (CC) procedure.

**Figure 3 metabolites-12-00817-f003:**
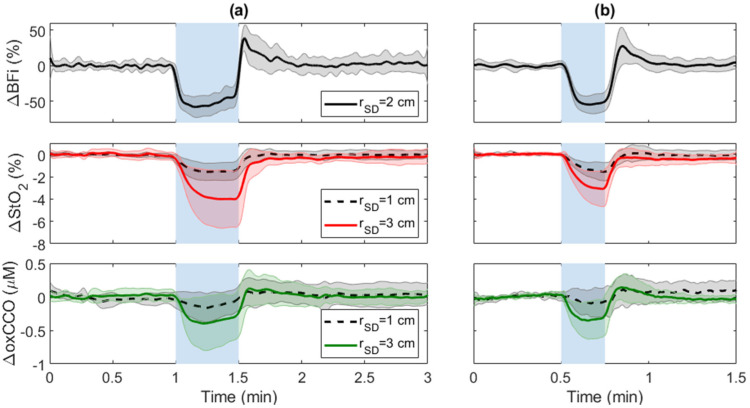
Average changes in StO_2_, oxCCO, and BFi in response to ipsilateral (**a**) 30-s CC (blue region between 1 and 1.5 min) and (**b**) 15-s CC (blue region between 0.5 and 0.75 min). Time courses were averaged across subjects, and shading surrounding each line represents the standard deviation.

**Figure 4 metabolites-12-00817-f004:**
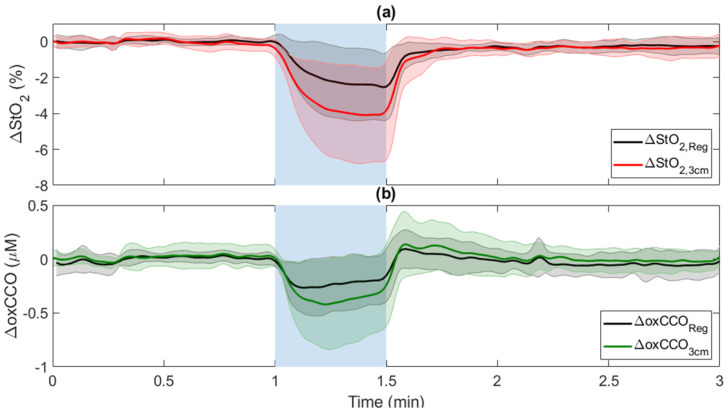
Regression analysis of (**a**) ΔStO_2_ and (**b**) ΔoxCCO in response to ipsilateral 30-s CC (indicated by blue region between 1 and 1.5 min). In both cases, the time course measured at *r*_SD_ = 1 cm was used as the regressor. Time courses were averaged across subjects, and shading surrounding each line represents the standard deviation.

**Figure 5 metabolites-12-00817-f005:**
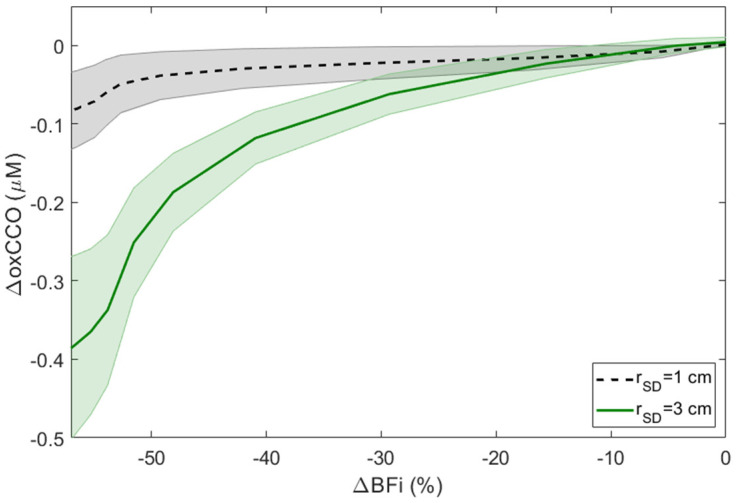
Relationship between reductions in BFi and corresponding changes in oxCCO. Results were averaged across subjects, and shading surrounding each line represents the standard deviation.

**Figure 6 metabolites-12-00817-f006:**
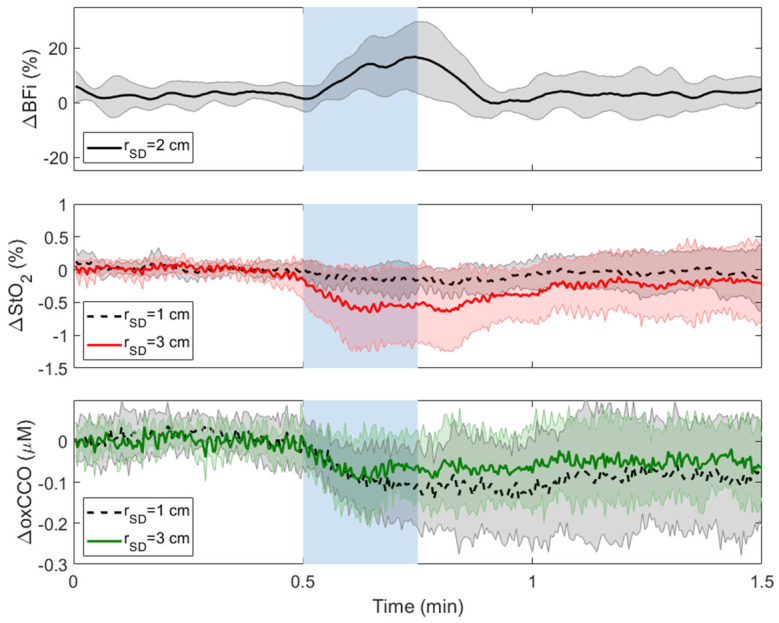
Average temporal change in BFi in response to contralateral 15-s CC, indicated by the blue region. Time courses were averaged across subjects, and shading surrounding each line represents the standard deviation.

**Figure 7 metabolites-12-00817-f007:**
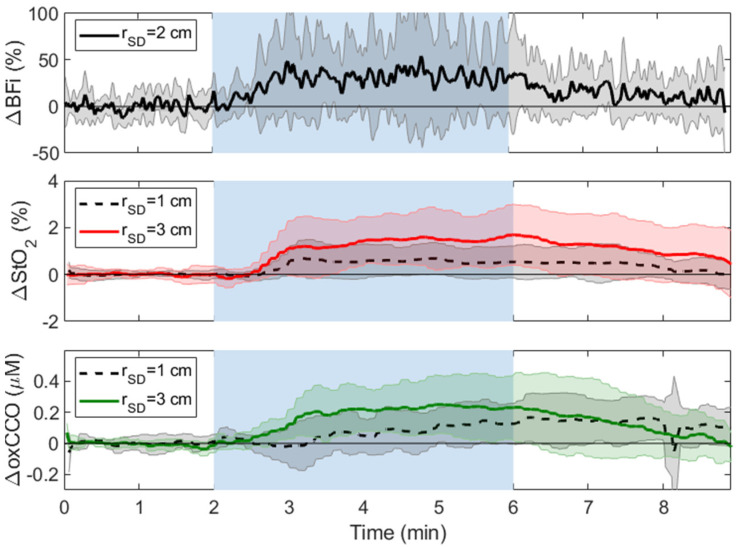
Average changes in BFi, StO_2_, and oxCCO in response to a 4-min hypercapnic challenge indicated by the blue shading. Time courses were averaged across subjects, and shading surrounding each line represents the standard deviation.

**Figure 8 metabolites-12-00817-f008:**
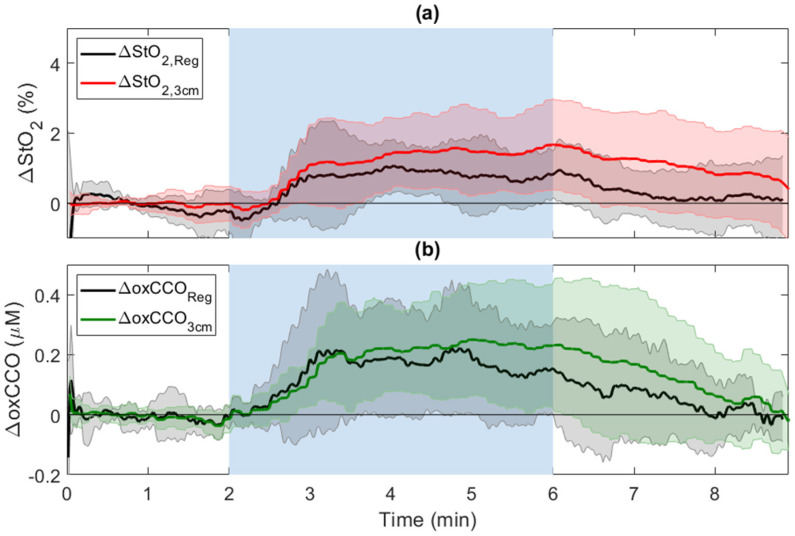
Regression analysis of (**a**) ΔStO_2_ and (**b**) oxCCO_3cm_ in response to hypercapnia (blue-shaded region).

**Table 1 metabolites-12-00817-t001:** Average oxygenation and metabolic responses for 30-s and 15-s CC.

CC	Ipsilateral					Contralateral	
Duration of CC (s)	30		Regression	15		15	
*r*_SD_ (cm)	1	3	–	1	3	1	3
ΔStO_2_(%)	−1.2 ± 0.7 ^▽^	−4 ± 2.2 ^▽,^*	−2.4 ± 1.9 ^▽^	−1 ± 0.5 ^▽^	−3.1 ± 1.1 ^▽^^,^*	−0.2 ± 0.2	−0.6 ± 0.6
ΔoxCCO (μM)	−0.06 ± 0.1	−0.4 ± 0.3 ^▽^^,^*	−0.21 ± 0.24 ^▽^	−0.07 ± 0.2	−0.3 ± 0.2 ^▽^^,^*	−0.12 ± 0.08	−0.1 ± 0.1

Carotid compression (CC), source-detector distance (*r*_SD_), tissue oxygen saturation (StO_2_), oxidation state of cytochrome c oxidase (oxCCO). * 3 cm vs. 1 cm, ^▽^ reduction vs. baseline.

**Table 2 metabolites-12-00817-t002:** Average hemodynamic and metabolic changes during and following hypercapnia.

	Hypercapnia (min 4–6)		Hypercapnia Regression	Post Hypercapnia (min 7–9)		Post Hypercapnia Regression
*r*_SD_ (cm)	1	3	-	1	3	-
ΔStO_2_ (%)	0.6 ± 0.6	1.5 ± 1.1 ^▽^^,^*	0.82 ± 0.75 ^▽^	0.3 ± 0.6	0.9 ± 1.3 ^▽^	0.03 ± 0.1 ⯁
ΔoxCCO (µM)	0.1 ± 0.1	0.22 ± 0.19 ^▽^^,^*	0.15 ± 0.11 ^▽^	0.14 ± 0.1 ^▽^	0.1 ± 0.1 ●	0.03 ± 0.7

Carotid compression (CC), source-detector distance (*r*_SD_), tissue oxygen saturation (StO_2_), oxidation state of cytochrome c oxidase (oxCCO). * 1 cm vs. 3 cm, ^▽^ change vs. baseline, ⯁ regression vs. 3 cm, ● post hypercapnia vs. hypercapnia.

## Data Availability

Data can be made available by contacting the authors. Because of the participant consent obtained as part of the recruitment process, it is not possible to make these data publicly available.

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
