# Peer review of "Assessing the Sensitivity of Multi-Distance Hyperspectral NIRS to Changes in the Oxidation State of Cytochrome C Oxidase in the Brain"

_metabolites, 2022, doi:10.3390/metabo12090817_

Round 1
Reviewer 1 Report
This paper shows an interesting work on hsNIRS measurements of oxCCO. However, the hypothesis in the abstract and introduction shoule be reconstructured for the results: This study demonstrated that oxCCO was less sensitive to extracerebral signals than StO2.
Reviewer 2 Report
Dear Author
The topic entitled "Assessing the sensitivity of multi-distance hyperspectral NIRS to changes in the oxidation state of cytochrome c oxidase in the brain" is interesting and practical. in general, the manuscript is well-structured. but it needs to be improved:
1) The obvious innovation of the research should be revealed
2)The results of present paper should be compared with other Conventional method (it will be better it is done in the form of table).
Reviewer 3 Report
The subject of the paper is appropriate for the publication in the journal Metabolites. The submitted paper deals with the utilization of hyperspectral (hs)NIR spectroscopy to brain monitoring using measurements of the redox state of cytochrome c oxidase (∆oxCCO). The data were acquired during carotid compression (CC) to reduce the blood flow in the brain and hypercapnia to increase the flow. The authors demonstrated that hsNIR measurements of ∆oxCCO is less sensitive to extracerebral signals than measurements of tissue oxygen saturation (StO2) and can be, in a principle, utilized in brain monitoring during cardiac and vascular surgery. The manuscript is written well, the paragraphs are logically ordered.
However, the following points should be explained or clarified before this manuscript will be published.
The sentence “In contrast to StO2, cytochrome c oxidase, a protein complex that serves as a terminal storehouse for electrons………….” (p. 2, l. 60-63) should be re-written, The term “storehouse” is not appropriate and also notion “directly related to ATP production…” is not correct.
What does the expression “across hundreds of wavelengths“ (p.2, l. 72) mean? Maybe it should be “hundreds of nanometers”.
It would be very useful to provide a value of the total concentration of CCO in the studied region of the brain during the experiments.
It will be better to assign the value of “mean” as 29 years (p.6, l. 225) and also information about males involved in the experiments is missing.
The extinction coefficient for CuA in CCO (at 830 nm) is quite low ca 2 mM-1.cm-1. If the concentration change of CCO is about 0.1 µM, thus corresponding change in the absorbance (if the path length is 1 cm) is only 0.0002. What is the reproducibility of the measurements of this small change in the absorbance?
How is the redox state of CCO, determined by the redox state of CuA, correlated with the catalytic activity of CCO? Can you discuss this point?
Only few grammatical, stylistic, and formal errors are present in the manuscript, e.g.:
- the title of the journal is missing in refs.7 and 35
Round 2
Reviewer 1 Report
It can be accepted now.
Reviewer 2 Report
Dear Authors
Thanks for your answers. now it can be accepted for publication